# Measuring Impulsivity in Greek Adults: Psychometric Properties of the Barratt Impulsiveness Scale (BIS-11) and Impulsive Behavior Scale (Short Version of UPPS-P)

**DOI:** 10.3390/brainsci11081007

**Published:** 2021-07-29

**Authors:** Marianna Tsatali, Despina Moraitou, Georgia Papantoniou, Evangelia Foutsitzi, Eleni Bonti, Georgios Kougioumtzis, Georgios Ntritsos, Maria Sofologi, Magda Tsolaki

**Affiliations:** 1Greek Association of Alzheimer’s Disease and Related Disorders (GAADRD), 54643 Thessaloniki, Greece; despinamorait@gmail.com (D.M.); gpapanto.22@gmail.com (G.P.); tsolakim1@gmail.com (M.T.); 2Center for Interdisciplinary Research and Innovation (CIRI—AUTh) Balkan Center, Laboratory of Neurodegenerative Diseases, Buildings A & B, GreeceLab of Psychology, Section of Experimental & Cognitive Psychology, School of Psychology, Aristotle University of Thessaloniki, 54124 Thessaloniki, Greece; 3Laboratory of Psychology, Section of Experimental & Cognitive Psychology, School of Psychology, Aristotle University of Thessaloniki, 54124 Thessaloniki, Greece; 4Laboratory of Psychology, Department of Early Childhood Education, School of Education, University of Ioannina, 45110 Ioannina, Greece; efoutsitzi@gmail.com (E.F.); msofolo@yahoo.gr (M.S.); 5Department of Psychology, School of Social Sciences, University of Ioannina, 45110 Ioannina, Greece; 6Department of Psychiatry, Faculty of Health Sciences, School of Medicine, Aristotle University of Thessaloniki, 54124 Thessaloniki, Greece; elina.bonti@gmail.com; 7Department of Education, School of Education, University of Nicosia, 1700 Nicosia, Cyprus; 8Department of Special Education (CEDU), United Arab Emirates University (UAEU), Al Ain 15551, United Arab Emirates; 9Department of Turkish and Modern Asian Studies, National and Kapodistrian University of Athens, 15772 Athens, Greece; georgetype@gmail.com; 10Department of Hygiene and Epidemiology, School of Medicine, University of Ioannina, 45110 Ioannina, Greece; gntritsos@uoi.gr; 11Department of Informatics and Telecommunications, School of Informatics and Telecommunications, University of Ioannina, 45110 Ioannina, Greece; 12Institute of Humanities and Social Sciences, University Research Centre of Ioannina (U.R.C.I.), 45110 Ioannina, Greece; 131st Department of Neurology, Medical School, Aristotle University of Thessaloniki, 54124 Thessaloniki, Greece

**Keywords:** adaptation study, Barratt Impulsiveness Scale (BIS-11), Impulsive Behavior Scale (short UPPS-P), impulsivity, validation study

## Abstract

Introduction: The aim of the present study was to validate the Barratt Impulsiveness Scale (BIS-11th version) scale as well as the short version of the Impulsive Behavior Scale (UPPS-P) in a population of Greek young adults. Secondly, we aimed at validating the BIS-11 in older adults. Methods: 167 (Group 1) university students completed the Greek version of the BIS-11 (BIS-11-G) and the UPPS (UPPS-P-G) scales. Additionally, BIS-11-G was also administered to 167 (Group 2) cognitively intact older adults, to identify whether it could be used to measure impulsivity in an older adult population. Results: Both scales had satisfactory internal reliability and test–retest reliability, as well as convergent validity in the young adult population. In regard to the factor structure, a principal component analysis (PCA) extracted two factors for the BIS-11-G in the young adult population and three factors in older adults, as well as three factors for the short UPPS-P-G in young adults. Conclusions: The BIS-11-G and the UPPS-P-G scales can be used to measure different aspects of impulsivity in the Greek population of different ages in research and clinical practice.

## 1. Introduction

According to Whiteside and Lynam [1], the construct of impulsivity—which is an essential dimension of personality—has a relatively clear definition compared to other psychological characteristics. This is likely due to the fact that the various tools measuring impulsivity are significantly associated with one another. Taking into account the aforementioned statement as well as the position of several theories that impulsivity is related to distinct personality traits and to neuropsychological functions, the present study aimed to test the psychometric properties of two psychometric tools measuring impulsivity in the Greek adult population: the Barratt Impulsiveness Scale (BIS) and the Impulsive Behavior Scale (UPPS-P). Both measures have been used in a wide range of genetic, neuropsychological and psychological studies in non-Greek populations [2].

The Barratt Impulsiveness Scale (BIS-11; 11th version; [3]) is one of the most commonly administered self-report measures for the assessment of impulsivity in both research and clinical settings. It provides a sum score and three sub-scores derived from its three subscales (the “attentional impulsiveness” subscale, the “motor impulsiveness” subscale and the “non-planning impulsiveness” subscale). The BIS-11 has been associated with objective, as well as self-reported, psychometric measures of various facets of impulsivity and was administered in the American population. 

However, there is large discrepancy across studies concerning the variance of the factor structure of BIS [4] in different cultural contexts; therefore, investigation of its structure in the Greek population is warranted. Specifically, according to studies conducted in Arabic [5], Brazilian [6] and Indian [7] adult populations, as well as German [8] adolescents, the BIS-11 was found to have a three-factor solution. Additionally, it was found that the BIS-11 had great fit in Japanese adults and adolescents [9]. The factors extracted in Italian adults were also identical with those in the original version (i.e., three-factor solution) [10]. Nevertheless, it is worth mentioning that the items in the Portuguese version [6], as well as in the Indian version (only of the three factors), did not follow the original factor’s loading, whereas the German version had slight differences from the original English version. On the contrary, studies including Norwegian neurological patients [11], as well as healthy adults, and also Italian [12] and Chinese adolescents [13] supported a two-factor structure, stating that the results should be interpreted with caution due to the moderate to low model fit. 

In order to explain the big conflict existing in numerous studies, Steinberg et al. [14] introduced the Barratt Impulsiveness Scale-Brief (BIS-Brief), which was found to be similarly valid compared to the original BIS-11, without the additional administration time. This is in line with the study of Reid et al. [15], who suggested that their results extracted a 12-item three factor structure, which includes almost half of the items of the original BIS-11 factor model. 

Finally, Vasconcelos et al. [2], in a review of the psychometric proprieties of BIS-11, stated that both the numbers and content factors assessed were unstable, which may lead to potential misinterpretations of the scores used to characterize impulsivity in clinical and non-clinical groups. They also commended new efforts to explore BIS-11 items’ internal organization rather than applying the initial model to calculate scores. Conclusively, due to the wide variation in the factor structure of this psychometric tool, the present study aimed to explore the BIS-11 items’ internal organization in the Greek population. 

The UPPS-P Impulsive Behavior Scale is also a widely accepted tool used to conceptualize the multivariable structure of impulsivity. Specifically, the UPPS-P Impulsive Behavior Scale has the following acronyms; U: negative urgency, P: premeditation (lack of), P: perseverance (lack of), S: sensation-seeking, P: positive urgency. It intends to identify various aspects of impulsivity and is universally accepted as a well-structured personality model [16]. According to the study by Whiteside, Lynam, Miller and Reynolds [16], the aforementioned personality facets are present in the most widely used personality models, and therefore, can capture the theoretical framework of personality. In regards to the factor structure of UPPS-P, the large corpus of literature has shown a clear three-factor solution, according to studies conducted in the Spanish [17], English [18], French [19] and Arabic [20] adult population.

### Aim of the Study

To date, according to our existing knowledge, there is no prior research data on the psychometric properties of both aforementioned scales in the Greek adult population. Therefore, this study aims to examine the psychometric properties (i.e., internal consistency reliability, test–retest reliability, factorial validity and convergent validity) of the Barratt Impulsiveness Scale (BIS-11) and the short version of the Impulsive Behavior Scale (UPPS-P), in order to accurately and reliably measure impulsivity in the Greek adult population.

Although both scales are widely used to measure the individual aspects of impulsivity, aforementioned studies have shown that the BIS factor structure is not stable across all different cultural adaptations. This is in contrast to UPPS-P, for which the literature has shown a more clear factor structure. To study the BIS-11 scale in more depth, we recruited a second group of participants from an older adult population to find possible differences between young adults and old adults and explore whether factor analysis could indicate the same three-factorial structure in young and old adult populations. Through this division of our sample into two age categories, we aim to further extrapolate the properties of the BIS-11 literature and therefore, expand its understanding. An additional vital reason for recruiting older adults is due to the necessity of determining which of the questionnaires used in the current study have better fit with facets of impulsivity in older adults. Investigating whether BIS-11 has a good fit in older adults will be useful for clinicians to detect those with increasing impulsivity, which is often associated with pathological conditions in ageing.

## 2. Methods

### 2.1. Participants

The young adults’ sample (Group 1) included 167 participants, who were undergraduate and postgraduate students from public and private Universities in Northern and Western Greece. The majority of the sample was female (young adult’s group, Group 1: 48 males and 119 females), whereas their mean age was 23.12 years (*SD* = 5.8) and their mean education years were 14.8 (*SD* = 1.33). Participants were recruited from researchers’ personal and professional contacts and through announcement boards in the Aristotle University of Thessaloniki, as well as the University of Ioannina. Therefore, the reason for the different sex ratio can be explained due to the fact that the participants of our study were mainly from humanistic departments, in which females are more common than males. The older adults’ sample (Group 2) consisted of 167 participants (46 males and 121 females) whose mean age was 71.07 (*SD* = 7.8), and their mean education years was 10.5 (*SD* = 4.3), as measured by schooling years. Older adults were recruited from the Greek Alzheimer’s Association where they had initially gone to be tested for their memory for precautionary reasons, as well as from older adult care centers. After they had undergone the official neurological and neuropsychological assessment, cognitively intact elders, as well as elders with subjective cognitive decline (SCI), who did not meet the criteria for mild cognitive impairment (MCI), were recruited in the present study at the same time just after their evaluation. The reason for recruiting more females than males is that more females come for evaluation in the Greek Alzheimer’s Association compared to males, probably because they worry more about their cognitive status.

### 2.2. Procedure

The participants took part in the study individually, on a volunteer basis. Initially, they read and signed the written informed consent prior to the psychometric tools’ administration, agreeing that the research team of this study could use their basic demographic information and also their scales’ total scores for research reasons. After that, they completed the demographic questionnaire and the two psychometric tools. The examination lasted about 30 min maximum. In order to measure the test–retest reliability of the BIS-11-G and the short UPPS-P-G scales, 25 participants of the initial Group 1 completed the scales one month after the first assessment. Additionally, 25 participants from Group 2 completed the BIS-11-G, as an attempt to investigate in more detail the observed differences across the studies in regards to its factor structure, as mentioned previously.

The examination lasted about 30 min maximum and involved a face-to-face assessment session in which the participants completed the demographic history as well as the scales that were administered. The study was approved by the ethics committee of the Greek Alzheimer Association (n 51A/10-10-2018). Before the administration of the scales for the study demands, we administered the BIS-11-G, as well as the short UPPS-P-G, to 25 university students in order to detect any difficulties in understanding or any objections that they might have had in regards to the content of the Greek versions of the scales. The pilot survey indicates that there was no problem with the translation of the scales, nor were there any objections regarding the understanding of the text or any refusal to participate in the study. The duration of the study was from September 2018 to November 2019, because the participants were not all recruited at the same time.

### 2.3. Instruments

#### 2.3.1. Barratt Impulsiveness Scale (BIS-11)

The BIS-11 (11th version) [10] is the latest version of the BIS (Patton et al., 1995) and is designed to measure the behavioral or personality trait of impulsivity. The tool consists of 30 items, and the participant is requested to answer each item on a 4-point Likert scale from 1 (rarely/never) to 4 (almost always/always). Although there is a pre-existing translation of the BIS-11 in Greek by Giotakos et al. [21], to our knowledge, until now, there has been nothing similar to the present study in the Greek population. Therefore, we adapted the translated version by Giotakos et al. from the beginning by doing a pilot study before we moved on with the validation study. To date, this is the first time the BIS-11 psychometric properties have been tested on Greek people, in both young and older adults, as there is no Greek BIS-11 validation study published. 

#### 2.3.2. Impulsive Behavior Scale (Short UPPS-P)

The UPPS-P scale was initially developed by Whiteside and Lynam [1] and is composed of 59 items. However, in the current study, we used the short version of the UPPS-P, because it takes less time to administer while maintaining its satisfactory psychometric properties [18] in comparison to the original form. The short UPPS-P is a 20-item version of the original UPPS-P scale designed from the five factor model of personality [22] to measure impulsive predispositions and, specifically, the five impulsive traits, mentioned as first-order factors, suggested by the structural model of Whiteside and Lynam [1], that is, (1) emotion-based rash action that includes negative and positive urgency, (2) sensation-seeking and (3) deficits in conscientiousness consisting of a lack of premeditation and a lack of perseverance.

Due to the lack of any short UPPS-P validation in the Greek population, we had initially conducted the translation process of the English version of the scale into Greek (Greek short UPPS-P; short UPPS-P-G) using the translation-back translation method [23]. Specifically, we translated the English version of the UPPS-P into Greek by two English–Greek bilinguals. The originally translated and the back-translated Greek versions were then compared for consistency, relevance and the meaning of the content. 

### 2.4. Data Analysis

To test the internal consistency of the Greek BIS-11 and short UPPS-P, Cronbach’s alpha coefficient (minimum acceptable value is 0.7) was calculated [24]. Additionally, intraclass correlation (ICC) has been increasingly employed to detect the scale’s test–retest reliability. Specifically, twenty-five (25) participants from each one of the two groups (Time 1), who were randomly chosen from the original sample, were administered the BIS-11-G a second time for test–retest reliability after a one month period (Time 2), whereas twenty-five (25) participants from Group 1 were also administered the UPPS-G after a one month period (Time 2). For measuring convergent validity, which is part of construct validity, we used the Pearson correlation coefficient to identify whether the two psychometric tools were significantly related, whereas their structural validity was tested by conducting exploratory factor analysis. In more detail, to test whether the BIS-G as well as the short UPPS-P-G maintained the same factor structure as their original English versions [1] (the short UPPS-P version from Cyders et al. [18]; BIS-11th form Patton et al. [3]), we conducted principal component analysis (PCA). The reason for using PCA is because it is the preferable method of controlling the structure of these tools at this stage, in which it is the first time these scales have been adapted to the Greek population. Regarding the application of the PCA method, criterion values proposed by Hu and Bentler [25] were used, in order to assess the fit of the model. Specifically, in order to extract the factor measures, the following criteria were met: the eigenvalues, which estimate the amount of variance subjected by each factor; the scree plot, which depicts different observable components; the Kaiser–Meyer–Olkin (KMO) measure for estimating the sampling adequacy (acceptable min score is 0.50), as well as Bartlett’s test of sphericity. Finally, a paired samples T-test was also used to investigate the possible differences between the two groups concerning the BIS-11-G mean scores. Statistical analyses were conducted by SPSS version 25 (IBM Corp., Armonk, NY, USA).

## 3. Results

### 3.1. Descriptive Statistics

The demographic data of the sample is presented in Table 1, whereas mean scores and standard deviations of the scales administered are placed in Table 2. 

### 3.2. BIS-11-G 

#### 3.2.1. Scale’s Validity

Structural validity: We conducted principal component analysis (PCA). Specifically, in order to extract the scale’s factors in both groups, PCA was conducted using Varimax rotation, because the factors of impulsivity were not found to be correlated. The minimum eigenvalue was set at 1 to speculate the extraction criterion. We also based the scree plot to extract the right number of factors.

Concerning the BIS-11-G in Group 1, PCA showed a two-factor structure that accounted for 37.96% of the total variance. The KMO of BIS-11-G in Group 1 of the present study was found to be 0.80 indicating a good sampling accuracy, whereas Bartlett’s test of sphericity was found to be significant *χ*^2^(300) = 1594.113, *p* < 0.001. Sixteen items loaded on the first factor, and nine items on the second factor (factor loading was ≥0.30).

The two factors were clearly identified. Specifically, the first factor labeled as “motor impulsiveness” (accounting for the 25.99% of the total variance), whereas the second factor, which was labeled as the “non-planning impulsiveness” (accounting for 11.97% of the total variance), also includes items from the factor called “attentional impulsiveness” in the original version of BIS. The loadings of the items on the factors for the BIS-11-G are placed in Table 3. 

Regarding the BIS-11-G in Group 2, a three-factor structure was found, which accounted for the 33.77% of the total variance. Τhe KMO of BIS-11-G was found to be 0.69, indicating a good sampling accuracy, whereas Bartlett’s test of sphericity was found to be significant, *χ*^2^(378) = 1162.125, *p* < 0.001. Ten items loaded on the first factor, labeled “non-planning impulsiveness” (accounting for 17.15% of the total variance); nine items were loaded to the second factor, labeled “motor planning impulsiveness” (accounting for 11.38% of the total variance), whereas six items loaded on the third factor, named “attentional impulsiveness” (accounting for 8.72% of the total variance) (factor loading was ≥0.40). The loadings of the items on the factors for the BIS-11-G in Group 2 are placed in Table 4.

#### 3.2.2. Scale’s Reliability

Internal consistency reliability: Cronbach’s alpha coefficient of the Greek BIS-11 for the 25 items was fairly high, especially for young adults (Group 1; Cronbach’s *α* = 0.871, Group 2; Cronbach’s *α* = 0.841), suggesting good internal consistency. As regards the Cronbach’s *α* coefficients for the specific factors (subscales) of the Greek BIS-11, in Group 1 were the following; *α* = 0.852, for factor 1; *α* = 0.841, for factor 2, whereas in Group 2 were the following; *α* = 0.795, for factor 1; *α* = 0.655 for factor 2; *α* = 0.640 for factor 3. The aforementioned coefficients were fairly high for Group 1 and marginally acceptable to high for Group 2. 

Test–retest reliability: A high degree of reliability was found between the Time 1 and Time 2 administration of the BIS-11-G in both groups. In more detail, concerning BIS-11-G in Group 1, the average measure of ICC was 0.906 with a 95% confidence interval from 0.675 to 0.947; *F*(25,25) = 10.677, *p* < 0.001), whereas in Group 2, the relevant ICC score was 0.946 with a 95% confidence interval from 0.812 to 0.984; *F*(25,25) = 18.481, *p* < 0.001).

### 3.3. UPPS-P-G

#### 3.3.1. Scale’s Validity

Structural validity: A PCA was conducted using the Varimax rotation method, in order to identify whether the short UPPS-P-G has the same factor structure in line with the original short UPPS-P scale. The minimum eigenvalue was set at 1 to speculate the extraction criterion. The KMO was found to be 0.85, which indicates good sampling accuracy. The Bartlett’s test of sphericity was found to be significant *χ*^2^(190) = 1422.988, *p* < 0.001. Initially, the factor analysis, taking into account the eigenvalue criterion, had shown a four-factor solution. However, the scree plot indicated a three-factor solution and, given that this solution agrees with the original model proposed by the authors, we decided to keep this factor structure of the short UPPS-P-G. The three-factor structure of the UPPS-P-G accounts for 55.7% of the total variance. Specifically, eight items loaded on the first factor, seven items on the second factor and five items on the third factor (factor loading was ≥0.45). The first factor was labeled “deficits in conscientiousness” (eigenvalue = 5.57; accounting for 27.85% of the total variance); the second factor was labeled the “emotion-based rash action” (eigenvalue = 3.43; accounting for 17.16% of the total variance), whereas the third factor was labeled “sensation-seeking” (eigenvalue = 2.14; accounting for 10.73% of the total variance). The loadings οf the items on the factors for the short UPPS-P-G are placed in Table 5. When comparing the UPPS-P-G with the original scale, no differences were found in their factor structure. The items load on each of the three factors properly, except for item 17, “I act ‘on impulse’.” which loads on the “sensation-seeking” factor in UPPS-P-G in contrast to the original UPPS-P in which it loads on the “emotion-based rash action” factor. 

#### 3.3.2. Scale’s Reliability

Internal consistency reliability: Cronbach’s alpha coefficient of the short UPPS-P-G sum score for the 20 items was fairly high (Cronbach’s *α* = 0.837), as well as those of its factors which were good to strong; Factor 1—“deficits in conscientiousness” (Cronbach’s *α* = 0.918), Factor 2—“emotion-based rash action” (Cronbach’s *α* = 0.799) and Factor 3—“sensation-seeking” (Cronbach’s *α* = 0.737), suggesting satisfactory internal consistency reliability.

Test–retest reliability: According to the results, a high degree of reliability was found in the short UPPS-P-G sum score completion between Time 1 and Time 2. The ICC value was 0.888 with a 95% confidence interval from 0.612 to 0.968; *F*(24,24) = 8.943, *p* < 0.001.

#### 3.3.3. Convergent Validity

Convergent validity, a main type of criterion validity, is measured to investigate whether one psychometric tool associates with others that measure the same attribute, that is, impulsivity in the current case. Specifically, Pearson’s correlation coefficient was conducted to identify whether the two scales were significantly correlated and, therefore, to identify whether their convergent validity is satisfactory for the Greek versions of the scales. According to the results, the BIS-11-G scores were positively moderately correlated with the short UPPS-P-G (*r* = 0.379, *p* < 0.001), indicating good convergent validity in the Greek population for both of the scales. 

## 4. Discussion

The present research was conducted to enrich the literature on the psychological characteristic of impulsivity by measuring the psychometric properties of the Barratt Impulsiveness Scale (BIS-11) and the Impulsive Behavior Scale (UPPS-P) in the Greek adult population, thereby enabling Greek clinicians and researchers to measure it in various research protocols and investigate its clinical utility. This endeavor is of significant importance, taking into account the multifaceted structure of impulsivity, as well as its coexistence in a large variety of disorders both in the psychology and psychiatry fields. Most broadly, due to the gap of adapted and/or validated psychometric tools for assessing impulsivity in the Greek population, the current findings provide strong support for using the BIS-11-G και short UPPS-G in this population. In more detail, the aforementioned self-reported scales have been found to have adequate psychometric properties, specifically internal consistency, test–retest reliability, factorial validity and convergent validity, and therefore can be used to measure the different aspects of impulsivity in the Greek adult population. 

In regards to the BIS-11-G factor validity, we had initially recruited two different groups consisted of young (Group 1) and old (Group 2) participants due to the fact that the BIS-11 three-factor structure is not stable and is variable across various countries’ adaptation studies, as mentioned previously. For that reason, we chose to test the BIS-11-G in two different populations (young and old adults) in order to identify whether the inconsistency found in previous studies reflects age differences. Furthermore, another reason for recruiting older adults’ population, is because it is of outmost importance to identify psychometric tools that can reliably and validly detect increased levels of impulsivity, which are strongly associated with pathological conditions in ageing. Impulsivity includes personality trait characteristics [26], which are measured by specific self-reported scales. However, it is noteworthy that, although personality has traditionally been viewed as a ‘stable’ characteristic, longitudinal research has clearly demonstrated large changes across the lifespan [26]. Consequently, self-reported scales, which can identify whether impulsivity trait and/or state levels [27] have increased in old adults’ population, are quite useful for clinicians, because they can be assumed to be early signs of pathological disturbances such as frontotemporal dementia, Parkinson’s disease, etc. [28,29].

Concerning the factorial validity of BIS-11-G, the results of the present study are in agreement with the aforementioned inconsistency, because in Group 1 (young participants) was extracted a two-factor solution, whereas the factor structure of the BIS-11-G in Group 2 (old adults) was in line with the factor structure of the original BIS giving a three-factor solution. Another discrepancy between the original BIS and the BIS-11-G was that, according to the factor loadings, only the 25 items, in both groups, were loaded to the factors instead of the 30 items loaded on the initial BIS. Conclusively, as mentioned by Vasconcelos et al. [30], many studies have failed to confirm the three-factor model, which is also in agreement with our study, at least for young adults, whereas the number of items belonging to the final BIS-11 was also lower in both groups of participants in our study. Previous findings showed that the factor of “non-planning impulsiveness” tends to be the most frequently observed compared to the other two factors, since it can be clearly extracted in studies with a BIS-11 two-factor structure, whereas “attentional impulsiveness” is considered to be the most unstable factor [3,30]. The aforementioned findings are in accordance with our results concerning the factor structure of BIS-11-G in Group 1, since the two factors extracted were “motor impulsiveness” and “non-planning impulsiveness”, enriched by some of the items that initially belonged to “attentional impulsivity”, which seems to be the less identifiable factor of BIS-11-G.

The BIS-11-G factor structure, which was extracted from PCA for Group 2, showed a three-factor solution, which is similar to what was initially suggested by the constructors of the scale [3]. This finding is also in line with many studies [5,7,8]. However, the loadings of the three-factor solution in the Greek BIS-11 were lower compared to the original version, indicating possible cross-cultural differences between the model fit of BIS-11-G with the original version, mainly in the older adults’ population.

In our study, we conducted PCA to validate the factors of the original scales and provide details about which model best fits the data. However, confirmatory factor analysis (CFA) is commonly used in validation studies in which the factorial validity is investigated. The reason why we used this kind of analysis is because this is the first time the versions of BIS-11 and UPPS-P are adapted in the Greek population, and, therefore, we assumed that it would be more accurate to use this type of factor analysis.

Cultural discrepancies concerning the factor structure of BIS also exist across different countries. In detail, according to Vasconcelos et al. [30], the differences in factor analysis are observed even in country samples that have similar cultural frames. Due to the fact that the construct of impulsivity is multifactorial and strongly affected by cultural variables, it is expected to have different factor loadings across the different BIS versions. However, the reason why the amount of variance accounted for in the BIS-11-G is low is still questionable.

Furthermore, it is of foremost importance that, in addition to the different factor solutions that were found between the two groups, the factor loadings in Group 1 were higher compared to those of Group 2. These findings possibly indicate age differences in the structure of the construct of impulsivity between young and older adults’ populations due to large changes of personality across the lifespan [26].

Concerning the reliability of BIS-11-G, both internal consistency, as well as test–retest reliability, were found to be acceptable. It is worth mentioning that the internal consistency of BIS-11-G in young adults was higher than in older adults. This finding is in line with the factor loadings, which were also found to be better in Group 1 compared to Group 2.

Moving to the UPPS-P-G, the three-factor model, extracted from the present study, was found to have an acceptable model fit, in accordance with the initial one proposed by Whiteside and Lynam [1], which is a theory-driven model that includes three distinct components associated with impulsive behavior. This factor structure is also in line with previous studies [31]. The only difference between UPPS-P-G with the original version of the scale is that item 17, which initially belonged to the “emotion-based rash action” subscale, is now loaded to the “sensation-seeking” factor, which is the third factor of the UPPS-P-G. To sum, UPPS-P-G has a satisfactory factorial validity, which is in almost complete fit to the original scale, and therefore can be very useful to measure specific domains of impulsivity, such as deficits in conscientiousness, emotion-based rash action and sensation-seeking in the Greek adult population.

According to reliability analyses, the Cronbach’s *α* coefficients of the UPPS-P-G indicate a satisfactory internal consistency reliability, similar to those reported in the initial study of Whiteside and Lynam [1]. Moreover, the test–retest reliability was also satisfactory, and therefore, it can be assumed that the short UPPS-P-G can be regarded as a reliable tool that can produce similar results under various administrative conditions.

## 5. Conclusions

Finally, the two scales are both valid and reliable tools to measure impulsivity in young adults, whereas BIS-11-G can also be used in an older adults population. Additionally, as was expected, they are theoretically related to each other [1], according to the statistically significant moderate positive relationship that was found between their sum scores. Notably, this study attempts to contribute to the development of studies in this field and thereby expand our knowledge about the construct of impulsivity. Widening the knowledge about impulsivity, this Greek adaptation study of BIS-11 and short UPPS-P can be regarded as a significant input in the relevant literature. Future study should examine these scales in various clinical samples and investigate their effect on specific psychiatric disorders.

## Figures and Tables

**Table 1 brainsci-11-01007-t001:** Demographic variables for all study’s participants.

Variables	Group 1 (n = 167)	Group 2 (n = 167)
Age (years)	23.1	71.07
S.D.	5.8	7.8
Males % (n)	28.7 (48)	27.5 (46)
Females % (n)	71.3 (119)	72.5 (121)
Education (years)	14–20	2–20
Mean	14.8	10.53
S.D.	1.33	4.31
Primary education (<6 years)	-	58 (34.7)
Secondary education (7–12 years)	8.4% (14)	58 (34.7)
University students % (n)	86.2% (144)	-
University graduates	-	50 (29.9)
Master students % (n)	5.4% (9)	1 (0.6)

**Table 2 brainsci-11-01007-t002:** Scales’ mean scores and SD.

	Mean Scores (SD) Group 1	Mean Scores (SD) Group 2
**BIS-11-G sum score**	57.28 (10.32)	50.40 (9.14)
Factor1	33.78 (8.05)	20.72 (5.93)
Factor2	20.14 (5.59)	15.89 (4.04)
Factor3	-	13.77 (3.67)
**UPPS-P-G sum score**	44.26 (9.42)	-
Factor1	15.71 (5.84)	-
Factor2	16.33 (4.52)	
Factor3	12.40 (3.58)	

**Table 3 brainsci-11-01007-t003:** Component loadings in the BIS-11-G in group 1 (2 factor components).

BIS Items	1	2
2. I do things without thinking.	0.636	
3 I make-up my mind quickly.	0.523	
4 I am happy-go-lucky.	0.562	
5 I don’t “pay attention.”	0.578	
11 I “squirm” at plays or lectures.	0.502	
14. I say things without thinking.	0.571	
16 I change jobs.	0.496	
17. I act “on impulse.”	0.770	
18 I get easily bored when solving thought problems.	0.391	
19. I act on the spur of the moment	0.717	
21 I change residences	0.482	
22 I buy things on impulse.	0.488	
24 I change hobbies.	0.660	
25 I spend or charge more than I earn.	0.444	
26 I often have extraneous thoughts when thinking.	0.357	
28 I am restless at the theater or lectures.	0.531	
1 I plan tasks carefully. (R)		0.655
7 I plan trips well ahead of time. (R)		0.708
8 I am self-controlled. (R)		0.608
9 I concentrate easily. (R)		0.491
10 I save regularly. (R)		0.582
12 I am a careful thinker. (R)		0.761
13 I plan for job security. (R)		0.573
20 I am a steady thinker. (R)		0.750
30 I am future oriented (R)		0.714

**Table 4 brainsci-11-01007-t004:** Component loadings in the BIS-11-G in group 2 (3 factor components).

BIS-11-G Items	1	2	3
1 I plan tasks carefully. (R)	0.447		
7 I plan trips well ahead of time. (R)	0.468		
8 I am self-controlled. (R)	0.517		
9 I concentrate easily. (R)	0.548		
10 I save regularly. (R)	0.649		
12 I am a careful thinker. (R)	0.769		
13 I plan for job security. (R)	0.592		
15 I like to think about complex problems. (R)	0.496		
20 I am a steady thinker. (R)	0.645		
30 I am future oriented (R)	0.677		
2 I do things without thinking.		0.547	
3 I make-up my mind quickly.		0.475	
5 I don’t “pay attention.”		0.418	
14 I say things without thinking.		0.496	
17 I act “on impulse.”		0.675	
19 I act on the spur of the moment.		0.517	
22 I buy things on impulse.		0.483	
24 I change hobbies.		0.410	
25 I spend or charge more than I earn.		0.438	
6 I have “racing” thoughts.			0.405
11 I “squirm” at plays or lectures.			0.688
18 I get easily bored when solving thought problems.			0.591
26 I often have extraneous thoughts when thinking.			0.595
28 I am restless at the theater or lectures.			0.745
29 I like puzzles. (R)			0.454

**Table 5 brainsci-11-01007-t005:** Component loadings in the short UPPS-P-G (3 factor components).

UPPS-P-G Items	1	2	3
1. I generally like to see things through to the end.	0.831		
2. My thinking is usually careful and purposeful.	0.810		
4. Unfinished tasks really bother me	0.741		
5. I like to stop and think things over before I do them.	0.811		
7. Once I get going on something I hate to stop.	0.746		
11 I “squirm” at plays or lectures.	0.799		
12 I am a careful thinker.	0.789		
19 I act on the spur of the moment	0.783		
3. When I am in great mood, I tend to get into situations that could cause me problems. (R)		0.665	
6 When I feel bad, I will often do things I later regret in order to make myself feel better now. (R)		0.660	
8. Sometimes when I feel bad, I can’t seem to stop what I am doing even though it is making me feel worse. (R)		0.676	
10. I tend to lose control when I am in a great mood (R)		0.489	
13. When I am upset I often act without thinking. (R)		0.642	
15 I like to think about complex problems. (R)		0.684	
20 I am a steady thinker. (R)		0.718	
9. I quite enjoy taking risks (R)			0.688
14. I welcome new and exciting experiences and sensations, even if they are a little frightening and unconventional (R)			0.673
16 I would like to learn to fly an airplane. (R)			0.740
17 I act “on impulse.” (R)			0.587
18 I get easily bored when solving thought problems. (R)			0.726

## Data Availability

The study did not report any data.

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
