# Peer review of "Measuring Impulsivity in Greek Adults: Psychometric Properties of the Barratt Impulsiveness Scale (BIS-11) and Impulsive Behavior Scale (Short Version of UPPS-P)"

_brainsci, 2021, doi:10.3390/brainsci11081007_

Round 1

Reviewer 1 Report

This is a very interesting study, and it definitely reflects a positive efforts into adjusting impulsivity-measuring scales for the Greek adult populations.

The study is well conducted, clearly presented, and the conclusions expressed derive from highly relevant findings in regards to the reliability and validity of the translated scales used in the Greek adult population.

However, there are a number of suggestions for improvement, which are listed below. The vast majority of them related to the incorrect and/or informal use of the English language. Thus, except for so many of the corrections listed below, the authors are kindly requested to proofread their manuscript very carefully with the help of a native English speaker before resubmission.

  • It appears as if the document submitted has tracked changes included in it. Please make sure to fix this error in the revised submission.
  • Abstract: some rewording is required here for further clarity. Examples: "for Greek adult populations", "in AN older adult population", "in THE young population", "...to measure different aspects of impulsivity in THE Greek ADULT population OF DIFFERENT AGES in research and clinical practice".
  • Abstract: Do the three factors for the short UPPS-P-G identified correspond to both populations (i.e., university students and older adults?)
  • line 43-44; "are associated significantly with one another"
  • lines 41-46: both terms, impulsivity and impulsiveness, are mentioned, without any explanation of any potential difference between the two, and indeed, without an actual definition of either been included, other than a comment of a "relatively clear" definition.
  • lines 59-60; "quite useful". Informal and inaccurate, please rephrase.
  • line 61: "conducted in" not "to".
  • line 62: "was found"
  • line 63: "as well as THE adolescent..."
  • line 72: "please rephrase "With purpose to explain..."
  • line 76: "...which includes almost half of the items..."
  • line 79: "unstable"
  • line 84: "...in THE Greek population."
  • line 86: "Specifically" not "In specific"
  • lines 86-88: are all the acronyms correct, considering that at least some of initials "P" are the same? How does one differentiate the corresponding spell outs of at least three acronyms are P?
  • line 89: "According to their study..." According to whose study?
  • line 91: "In regards TO..." or "Regarding the..."
  • lines 93-94; "...studies conducted in...."
  • line 97: "... in THE Greek adult population." Please correct this throughout the manuscript as it is a constant mistake.
  • line 122 and 126: men and women, or males and females?
  • line 126-127: "...their mean education years were"
  • line 128 and 148: "Greek Alzheimer's Association"
  • line 131: "...who did not meet the criteria for..."
  • line 142: "...from Group 2..."
  • line 151: "they might have had in regards to the"
  • line 155: "were not all recruited at the same time"
  • lines 161-163: It is unclear why the translated version of the BIS-11 by Orestis Giotakos (2003) was not used in this study.
  • line 167: "tested in the Greek..."
  • line 306: "...in THE Greek adult population,..."
  • line 312: "...in THE Greek population..."
  • line 318: "In regards to...
  • line 323: "... the inconsistency... reflects age differences".
  • line 333: "... Parkinson's disease, etc."
  • Is is unclear why the Limitations are used at the very end of the manuscript as a separate section. It may be preferable to include a paragraph of the limitations within the Discussion section, and only present the Conclusions of this study at the very end as a "take home" message. Including the limitations at the very end of an extensive manuscript effort can only but leave a bittersweet taste to the reader.

Author Response

First of all, we would like to thank you for your thoughtful and constructive comments. We have been working on the revisions and we believe that the quality of our manuscript has greatly improved with the incorporation of your comments. In the following document you will find our answers to your comments.

Sincerely,

The authors

However, there are a number of suggestions for improvement, which are listed below. The vast majority of them related to the incorrect and/or informal use of the English language. Thus, except for so many of the corrections listed below, the authors are kindly requested to proofread their manuscript very carefully with the help of a native English speaker before resubmission.

The manuscript has undergone official English Editing

It appears as if the document submitted has tracked changes included in it. Please make sure to fix this error in the revised submission.

Done

Abstract: some rewording is required here for further clarity. Examples: "for Greek adult populations", "in AN older adult population", "in THE young population", "...to measure different aspects of impulsivity in THE Greek ADULT population OF DIFFERENT AGES in research and clinical practice".

Done.

Abstract: Do the three factors for the short UPPS-P-G identified correspond to both populations (i.e., university students and older adults?)

Reviewer 1 is right, this is a significant issue. However, we did not recruit this scale in older adults, because we already had their BIS scores. We have now made it more clear in Abstract (please see the yellow font)

line 43-44; "are associated significantly with one another"

Done

lines 41-46: both terms, impulsivity and impulsiveness, are mentioned, without any explanation of any potential difference between the two, and indeed, without an actual definition of either been included, other than a comment of a "relatively clear" definition.

Reviewer 1 is right. We have now deleted the term impulsiveness, because the term is impulsivity.

lines 59-60; "quite useful". Informal and inaccurate, please rephrase.

The term has been replaced with ‘suitable’

line 61: "conducted in" not "to".

Done

line 62: "was found"

Done

line 63: "as well as THE adolescent..."

Done

line 72: "please rephrase "With purpose to explain..."

It has been replaced with ‘ In order to ‘

line 76: "...which includes almost half of the items..."

Done

line 79: "unstable"

Done

line 84: "...in THE Greek population."

Done

line 86: "Specifically" not "In specific"

Done

lines 86-88: are all the acronyms correct, considering that at least some of initials "P" are the same? How does one differentiate the corresponding spell outs of at least three acronyms are P?

Despite that Reviewer 1 is right, the P are the same . Here we state the reference link http://www.impulsivity.org/measurement/UPPS_P

line 89: "According to their study..." According to whose study?

  1. Whiteside, S. P.; Lynam, D. R.; Miller, J. D.; Reynolds (2005). It has been now mentioned.

line 91: "In regards TO..." or "Regarding the..."

Done

lines 93-94; "...studies conducted in...."

Done

line 97: "... in THE Greek adult population." Please correct this throughout the manuscript as it is a constant mistake.

Done (the correction has been made throughout the text)

line 122 and 126: men and women, or males and females?

Males and Females

line 126-127: "...their mean education years were"

Done

line 128 and 148: "Greek Alzheimer's Association"

Done

line 131: "...who did not meet the criteria for..."

Done

line 142: "...from Group 2..."

Done

line 151: "they might have had in regards to the"

Done

line 155: "were not all recruited at the same time"

Done

lines 161-163: It is unclear why the translated version of the BIS-11 by Orestis Giotakos (2003) was not used in this study.

Thank you very much for this comment. We used the translated version by Orestis Giotakos. The back translation process was conducted in the UPPS-P (we have now made it clear)

line 167: "tested in the Greek..."

Done

line 306: "...in THE Greek adult population,..."

Done

line 312: "...in THE Greek population..."

Done

line 318: "In regards to...

Done

line 323: "... the inconsistency... reflects age differences".

Done

line 333: "... Parkinson's disease, etc."

Is is unclear why the Limitations are used at the very end of the manuscript as a separate section. It may be preferable to include a paragraph of the limitations within the Discussion section, and only present the Conclusions of this study at the very end as a "take home" message. Including the limitations at the very end of an extensive manuscript effort can only but leave a bittersweet taste to the reader.

We have now moved the limitations into the Discussion (please see the yellow font)

Reviewer 2 Report

It would be interesting to adecuately justify the sex ratio, both in young and elderly sample.

Author Response

First of all, we would like to thank you for your thoughtful and constructive comments. We have been working on the revisions and we believe that the quality of our manuscript has greatly improved with the incorporation of your comments. In the following document you will find our answers to your comments.

Sincerely,

The authors

It would be interesting to adecuately justify the sex ratio, both in young and elderly sample.

Please see the paragraph 2.1 Participants